

# Convolutional neural networks to automate the screening of malaria in low-resource countries

Oliver S. Zhao[1], Nikhil Kolluri[2], Anagata Anand[1], Nicholas Chu[2], Ravali Bhavaraju[1], Aditya Ojha[2], Sandhya Tiku[1], Dat Nguyen[1], Ryan Chen[1], Adriane Morales[1], Deepti Valliappan[1], Juhi P. Patel[3] and Kevin Nguyen[3]

[1] Department of Biomedical Engineering, The University of Texas at Austin, Austin, TX, United States of America
[2] Department of Electrical & Computer Engineering, The University of Texas at Austin, Austin, TX, United States of America
[3] Department of Psychology, The University of Texas at Austin, Austin, TX, United States of America

## ABSTRACT

Malaria is an infectious disease caused by *Plasmodium* parasites, transmitted through mosquito bites. Symptoms include fever, headache, and vomiting, and in severe cases, seizures and coma. The World Health Organization reports that there were 228 million cases and 405,000 deaths in 2018, with Africa representing 93% of total cases and 94% of total deaths. Rapid diagnosis and subsequent treatment are the most effective means to mitigate the progression into serious symptoms. However, many fatal cases have been attributed to poor access to healthcare resources for malaria screenings. In these low-resource settings, the use of light microscopy on a thin blood smear with Giemsa stain is used to examine the severity of infection, requiring tedious and manual counting by a trained technician. To address the malaria endemic in Africa and its coexisting socioeconomic constraints, we propose an automated, mobile phone-based screening process that takes advantage of already existing resources. Through the use of convolutional neural networks (CNNs), we utilize a SSD multibox object detection architecture that rapidly processes thin blood smears acquired via light microscopy to isolate images of individual red blood cells with 90.4% average precision. Then we implement a FSRCNN model that upscales $32 \times 32$ low-resolution images to $128 \times 128$ high-resolution images with a PSNR of 30.2, compared to a baseline PSNR of 24.2 through traditional bicubic interpolation. Lastly, we utilize a modified VGG16 CNN that classifies red blood cells as either infected or uninfected with an accuracy of 96.5% in a balanced class dataset. These sequential models create a streamlined screening platform, giving the healthcare provider the number of malaria-infected red blood cells in a given sample. Our deep learning platform is efficient enough to operate exclusively on low-tier smartphone hardware, eliminating the need for high-speed internet connection.

Corresponding author
Oliver S. Zhao,
oliver.zhao@utexas.edu

## INTRODUCTION

### Malaria in developing countries

Malaria is an infectious disease caused by *Plasmodium* parasites, which are transmitted through female mosquito bites. *P. falciparum* is the most common and the deadliest human malaria parasite in Africa, accounting for nearly all fatal cases in Sub-Saharan Africa (*WHO, 2019*; *McKenzie et al., 2008*; *Makanjuola & Taylor-Robinson, 2020*). Typical symptoms include fever, malaise, headaches, and vomiting, and in severe cases, seizures and coma. The World Health Organization (WHO) reports that in 2018, there were 228 million cases and 405,000 deaths globally. Africa represents 93% of total cases and 94% of total deaths (*WHO, 2019*). The most vulnerable group of infected individuals are children under the age of five, where 67% of malaria deaths occur. The WHO suggests that rapid diagnosis and subsequent treatment are the most effective means to mitigate the progression into serious symptoms. However, less than 29% of children under the age of five in sub-Saharan Africa receive antimalarial drug treatment (*WHO, 2019*), despite this demographic being at the greatest risk (*Ricci, 2012*). The WHO cites that significant factors driving this statistic are poor access to healthcare and ignorance of malaria symptoms (*WHO, 2019*).

Malaria can be diagnosed based on clinical symptoms, although the Center for Disease Control (CDC) always recommends confirming the diagnosis with a laboratory test (*CDC, 2020*). Laboratory tests can include the use of PCR to identify the specific strain of *Plasmodium* in a confirmed malaria case (*Hong et al., 2013*), antigen detection kits to detect *Plasmodium*-derived antigens (*Polpanich et al., 2007*; *Khan et al., 2010*), and serology tests such as ELISA to detect antibodies targeting malaria parasites (*Murungi et al., 2019*). These methods are expensive and often infeasible to implement in low-resource settings due to the required equipment and use of trained technicians (*CDC, 2020*). In low-resource settings, the use of light microscopy on a thin or thick blood smear with Giemsa stain is often used to confirm the presence of malaria parasites (*Charpentier et al., 2020*). Infection severity is frequently measured through the percentage of red blood cells infected with malaria parasites, also known as percent parasitemia or parasitemia burden. However, the diagnostic accuracy of using Giemsa-strained thin blood smears depends heavily on the level of expertise in the technician, who must manually classify and count the number of malaria-infected red blood cells. This results in significant inter-observer variability due to the different levels of expertise in technicians in low-resource settings, who often have to learn other tasks and cannot be adequately trained for this specific task as a result (*Billo et al., 2013*; *Bowers et al., 2009*). For example, one study in Nigeria found that while both health providers and community members are familiar with malaria tests, there has been significant concern with the reliability of test results due to technician incompetency (*Ezeoke et al., 2012*). Meanwhile, microscopy-based diagnosis of malaria at primary health care facilities in Tanzania had a sensitivity of 74.5% and specificity of 59.0%, also indicating that technicians may not have proper training (*Ngasala et al., 2012*). A study in Angola also made similar conclusions that there is inadequate training for technicians involved in microscopy-based diagnosis of malaria (*Nazar-Pembele, Rojas & ngel Nez, 2016*).

**Table 1** **Previous attempts by other research groups to classify infected red blood cells.** A significant number of groups used their own datasets, while other groups used the NIH dataset.

| Source | Accuracy | Sensitivity | Specificity | Dataset |
|---|---|---|---|---|
| Ross et al. (2006) | 73.0 | 85.0 | NR | Private |
| Das et al. (2013) | 93.2 | 94.0 | 87.9 | Private |
| Adi et al. (2016) | 87.1 | NR | NR | Private |
| Liang et al. (2017) | 97.3 | 96.9 | 97.8 | NIH |
| Dong et al. (2017) | 98.1 | 97.3 | 98.7 | Private |
| Peñas, Rivera & Naval Jr (2017) | 92.4 | 95.2 | 84.7 | Private |
| Gopakumar et al. (2017) | 97.7 | NR | NR | Private |
| Rajaraman et al. (2018) | 98.6 | 98.1 | 99.2 | NIH |
| Rahman et al. (2019) | 97.7 | 97.4 | 97.9 | NIH |
| Rajaraman, Jaeger & Antani (2019) | 99.5 | NR | NR | NIH |

**Notes.**
NR, not reported.

## Use of machine learning in clinical applications and malaria screening

The use of machine learning methods, particularly neural networks, is rapidly growing in many areas of clinical application. The two primary applications are involved with either segmentation or classification in clinical images (*Shen, Wu & Suk, 2017*; *Anwar et al., 2018*; *Litjens et al., 2017*) or histological images (*Kan, 2017*; *Wang et al., 2019*). In particular, the use of machine learning to diagnose malaria is of interest, where various classification models were developed by several groups to determine whether a red blood cell is infected or uninfected, as shown in Table 1.

To address the severe malaria endemic in Africa and its related issues with medical resources and clinical expertise, we propose a multi-step automated screening process that takes advantage of readily available resources in low-income settings. Through the use of convolutional neural networks (CNNs), we utilize a $300 \times 300$ Single Shot MultiBox Detector (SSD300) multibox model for object detection (*Liu et al., 2015*) that rapidly processes Giemsa-stained thin blood smears acquired from basic light microscopy in order isolate images of individual red blood cells. Then we implement a separate FSRCNN image resolution upscaling model to raise the low-resolution images of $32 \times 32$ pixels to $128 \times 128$ pixels (*Dong, Loy & Tang, 2016*). The Fast Super-Resolution CNN (FSRCNN) model is only utilized if the images of individual red blood cells are of insufficient resolution due to the possible use of low-end cameras to acquire the thin blood smear images. Lastly, we utilize a variant of a VGG16 CNN that classifies every red blood cell as either infected or uninfected. These sequential models create a streamlined mechanism from which our screening platform takes in thin blood smear images as inputs to provide the healthcare provider with the number of infected red blood cells and parasitemia burden in a given sample. Taking advantage of the prevalent availability of low-end smartphones in the African continent, our deep learning platform is lean and efficient enough to operate exclusively on the smartphone hardware, eliminating the need for high-speed internet access to transmit image information into a cloud-based neural network model.

## METHODS

### Dataset and computing platform

Two datasets from different sources were used: (1) NIH malaria dataset and (2) Broad Institute malaria dataset. The publicly available NIH malaria dataset was acquired from the Lister Hill National Center for Biomedical Communications (LHNCBC) at the National Library of Medicine (NLM) located at https://lhncbc.nlm.nih.gov/publication/pub9932, which contains 27,588 labeled and segmented cell images acquired from Giemsa-stained thin blood smear slides. The dataset contains equal instances of uninfected red blood cells and *P. falciparium*-infected red blood cells derived from 150 *P. falciparium*-infected individuals and 50 uninfected individuals. Meanwhile, the Broad Institute dataset contains 1,364 blood smear images with 80,000 individually labeled blood cells that are either uninfected or infected with *P. vivax*, found at https://data.broadinstitute.org/bbbc/BBBC041/. In the Broad Institute dataset, only about 5% of the red blood cells are infected. All infected red blood cells in the NIH dataset are infected with *P. falciparum*, while all infected red blood cells in the Broad Institute dataset are infected with *P. vivax*.

The Google Cloud Platform (Google LLC, Mountain View, CA) was utilized for acquiring the bulk of experimental data from training different variations of the neural network models. Two Google Cloud Platform machine configurations were used: (1) N1 high memory machine with 8 vCPU and 52 GB memory with 1 Nvidia Tesla V100 GPU for experiments on partial datasets or (2) N1 high memory machine with 16 vCPU with 104 GB memory and 2 Nvidia Tesla V100 GPUs for experiments on full datasets. A boot disk with a Deep Learning on Linux operating system with the GPU Optimized Debian m32 (with CUDA 10.0) version was used to run all software on the Google Cloud Platform. In addition, the free online Google Colab interface with a T4 GPU was used for rapid code write-up and subsequent preliminary testing.

### Neural network performance metrics

In all neural network models used for classification and resolution enhancement, five-fold cross-validation was performed to report the mean and standard deviation of the model performance. The cross-validation groups were randomly split and distributed evenly among the five groups, with the same set of cross-validation groups used to test different model variants in a given experiment. Positive and negative samples were defined as infected and uninfected red blood cells, respectively. Some experiments did not utilize the full dataset, instead using a randomly selected subset of the dataset to reduce computational burden.

The object detection model performance was measured through average precision and average recall across different conditions, such as the intersection over union (IoU) values, image sizes, and maximum number of detections. The IoU values indicate the degree of overlap between the ground truth and predicted bounding boxes, with a high IoU indicating high overlap. The following metrics were measured in the malaria classification model: classification accuracy, sensitivity, specificity, area under the curve (AUC), F1-score, and Matthews correlation coefficient (MCC). The MCC is equivalent to the Phi coefficient, and is useful for evaluating imbalanced datasets such as the Broad Institute dataset (*Chicco &*

*Jurman, 2020*). While average precision and average recall are performance metrics used to describe object detection, we note that average precision corresponds to positive predictive value and average recall corresponds to sensitivity. The image upscaling model measured the mean squared error (MSE) and peak signal-to-noise ratio (PSNR) to examine the quality of the image upscaling output. Bicubic interpolation was used as the baseline for measuring comparing the performance of the CNN-based resolution upscaling model. The training and testing code and results are publicly available on a Github repository at https://github.com/oliver29063/MalariaDiagnosis.

## Development of object detection model

SSD300 (*Liu et al., 2015*) was trained to detect both infected and uninfected red blood cells from the thin blood smear images in the Broad Institute dataset. Because each red blood cell will be classified by the VGG16 classification model in later steps, the object detection model was not trained to distinguish between the two blood cell classes. The object detection model served primarily as a proof-of-concept to show that the mobile platform can sequentially run the object detection, resolution enhancement, and cell classification models in tandem. Consequently, the SSD300 model was not heavily fine-tuned to maximize performance. The final SSD300 model was trained with an RMSProp optimizer (*Ruder, 2016*) with a learning rate of 0.004. The batch size was 24 and the training process was run for 60,000 steps. All input images were scaled down via bilinear interpolation to the required 300 ×300 image size before entering the object detection model. The outputted thresholds from the 300 × 300 images were then rescaled to provide the original box coordinates of each individual red blood cell to isolate cropped images of each individual red blood cell.

## Development of the image classification CNN

All input images of the individual red blood cells from the NIH dataset were scaled to 128 × 128 resolution. In order to expand the number of hyperparameters examined, the CNN model was developed through sequential hyperparameter tuning rather than a traditional grid searchd or random search. First, the feature extraction architecture was optimized before developing the classification architecture. Then, hyperparameters involved with the training of the model—such as the optimizer, learning rate, and batch size—were fine-tuned to give the final model. All experiments with the image classification CNN were performed on a subset of 10,000 randomly selected images to reduce computational burden. After the final classification CNN was developed, the optimized hyperparameters were used to train on the entire dataset of 27,558 images to provide an accurate representation of the model performance.

### Fine-tuning the feature extraction architecture

During the fine-tuning of the feature extraction architecture, the following conditions were maintained for all experiments: (1) feature extraction layers were succeeded with two fully connected dense layers containing 512 nodes each with rectified linear unit (ReLU) activation functions and 50% dropout, and (2) an Adam optimizer with a learning rate of $10^{-6}$ and batch size of 64 was used. The following pre-trained CNN architectures with weights initialized from the ImageNet dataset were used: ResNet50V2, VGG16, VGG19,

InceptionV2, Xception, InceptionResNetV2, DenseNet121, and MobileNetV2. VGG16 and VGG19 are traditional deep CNNs (*Simonyan & Zisserman, 2015*), while ResNet50V2 uses residual connections to allow for deeper convolution layers (*He, Zhang & Shaoqing Ren and, 2016*). Other architectures such as Xception (*Chollet, 2016*), InceptionV2 (*Szegedy et al., 2014*), InceptionResNetV2 (*Szegedy et al., 2016*), MobileNetV2 (*Howard et al., 2017*), and DenseNet121 (*Huang et al., 2016*), build upon the use of residual connections. It is also worthwhile to note that MobileNetV2 is designed specifically for mobile phone use, sacrificing accuracy for the sake of speed. The top-performing model was chosen based on its overall accuracy and AUC. In the event of having similarly performing models, the model with the fewest parameters was selected to maximize model efficiency.

### Fine-tuning the classification architecture

The number of nodes in each of the two fully connected dense layers was tested with 128, 256, 512, and 1024 nodes each, with the set of dense nodes that resulted in the highest accuracy and convergence speed chosen. Then, the following dropout rates were examined: 25%, 50%, and 75%. The dropout rate resulting in the highest convergence speed and lowest testing loss was chosen. Lastly, the ReLU and Tanh activation functions were examined. When the given hyperparameter had yet to be fine-tuned, the experiments contained the following conditions: (1) 512 nodes in both dense layers, (2) 50% dropout, and (3) ReLU activation functions.

### Optimizing the learning conditions

The following optimizers were examined: stochastic gradient descent (SGD) with Nesterov momentum , Adam, RMSProp, AdaMax, and Nadam (*Kingma & Ba, 2014*; *Ruder, 2016*). The following learning rates were tested: $10^{-6}$, $10^{-5}$, $10^{-4}$, and $10^{-3}$. Graphical results have not been shown for learning rates that failed to train the model, although tabular results are available on the Github repository. The optimal learning rates were selected from each optimizer. Then, the performances of each optimizer were compared with the best optimizer chosen on the following three criteria: (1) final testing accuracy, (2) final testing loss, and (3) rate of convergence.

## Development of CNN-based image resolution upscaler

The FSRCNN model was developed in 2016 as an improvement over the previous SRCNN model introduced in 2014 (*Dong, Loy & Tang, 2016*; *Dong et al., 2014*). In short, the FSRCNN model performs feature extraction and shrinks a high dimensional feature map into a low dimensional feature map. Then a series of mapping layers process the features before the low dimensional feature map expands back to the high dimensional feature map. Finally, a deconvolution layer generates the high-resolution images. Consequently, the three main hyperparameters are: (1) number of mapping layers, (2) the dimension of the high feature map, and (3) the dimension of the low feature map. Consequently, we tested the FSRCNN using 2–4 mapping layers, 48 or 56 filters for high dimensional features, and 12 or 16 filters for low dimensional features.

In addition, we created two separate train and test sets to evaluate the effectiveness of the FSRCNN model: (1) FSRCNN-derived high-resolution train and test sets and (2) bicubic

**Table 2 SSD300 performance metrics.** Average precision (AP) and average recall (AR) across different IoUs, area sizes, and maximum number of detections. Top performing conditions for maximizing average precision and recall are bolded.

| Metric Type | IoU | Area size | Maximum detections | Performance |
|---|---|---|---|---|
| Average Precision (AP) | 0.50:0.95 | all | 100 | AP = 0.436 |
| **Average Precision (AP)** | **0.50** | **all** | **100** | **AP = 0.904** |
| Average Precision (AP) | 0.75 | all | 100 | AP = 0.491 |
| Average Precision (AP) | 0.50:0.95 | small | 100 | AP = −1.00 |
| Average Precision (AP) | 0.50:0.95 | medium | 100 | AP = 0.082 |
| Average Precision (AP) | 0.50:0.95 | large | 100 | AP = 0.440 |
| Average Recall (AR) | 0.50:0.95 | all | 1 | AR = 0.114 |
| Average Recall (AR) | 0.50:0.95 | all | 10 | AR = 0.295 |
| **Average Recall (AR)** | **0.50:0.95** | **all** | **100** | **AR = 0.639** |
| Average Recall (AR) | 0.50:0.95 | small | 100 | AR = −1.00 |
| Average Recall (AR) | 0.50:0.95 | medium | 100 | AR = 0.144 |
| Average Recall (AR) | 0.50:0.95 | large | 100 | AR = 0.605 |

interpolated high-resolution train and test sets. These train and test sets were then used to train and validate the final malaria classification model to examine how the differences in image quality impact the effectiveness of the classification CNN. Five-fold cross-validation with the full NIH dataset was used in these evaluations.

## Implementation of tensorflow lite android platform

TensorFlow Lite is an open-source platform focused on on-device model inference (*Abadi et al., 2015*). Unlike previously reported studies that utilize phone apps for model prediction (*Rajaraman, Jaeger & Antani, 2019*), this allows the models to run directly on the Android-based smartphones rather than relying on cloud-based computing resources. While all models were developed and trained with the TensorFlow and Keras packages, the final model deployments are subsequently converted into a .tflite file that allows the models to be run on the TensorFlow Lite package.

## RESULTS

### Red blood cell object detection model

The SSD300 object detection model trained on the Broad Institute dataset was able to detect the presence of red blood cells with an average precision of 90.4% when the IoU is 0.50 for all area sizes with 100 maximum detections, while the average recall was 63.9% at an IoU of 0.50:0.95 for all area sizes with 100 maximum detections, as shown in Table 2. We see that the model had high precision, but relatively poor recall. In Fig. 1 we see an example of the bounding boxes and confidence levels of detected red blood cells from a sample image from the Broad Institute dataset.

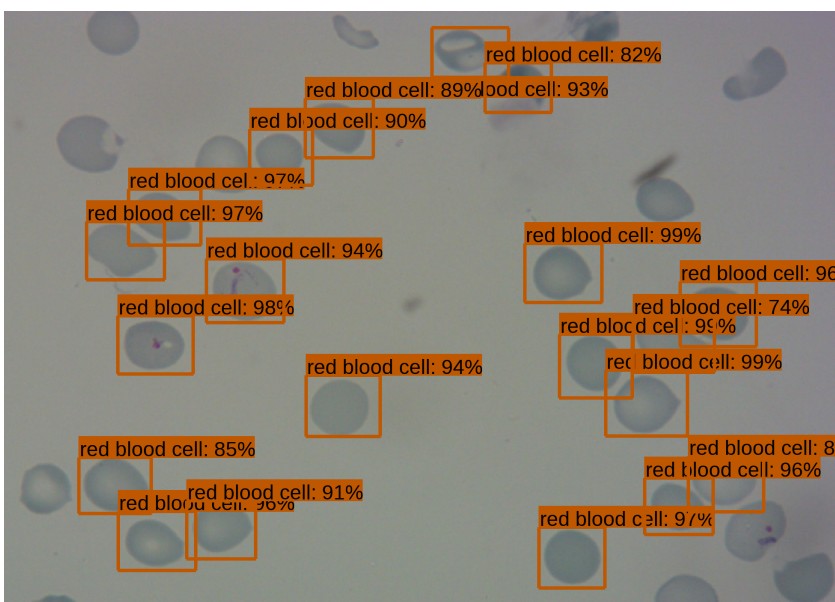

**Figure 1** **Sample image of Broad Institute dataset with object detection model outputs, such as bounding boxes and confidence thresholds.**

**Table 3** **Transfer learning performance metrics (mean ± std).** The partial NIH malaria dataset size contained 10,000 images with dense nodes set to 512 with ReLU activation functions. Adam optimizer with a learning rate of $10^{-6}$ and batch size of 64 was used.

| Model | Accuracy | Sensitivity | Specificity | AUC | F1 | MCC |
|---|---|---|---|---|---|---|
| ResNet50V2 | $0.938 \pm 0.009$ | $0.935 \pm 0.012$ | $0.940 \pm 0.010$ | $0.982 \pm 0.003$ | $0.935 \pm 0.012$ | $0.940 \pm 0.014$ |
| VGG16 | $0.960 \pm 0.003$ | $0.956 \pm 0.014$ | $0.964 \pm 0.010$ | $0.992 \pm 0.002$ | $0.956 \pm 0.014$ | $0.964 \pm 0.010$ |
| VGG19 | $0.959 \pm 0.004$ | $0.956 \pm 0.009$ | $0.963 \pm 0.010$ | $0.991 \pm 0.001$ | $0.955 \pm 0.009$ | $0.963 \pm 0.011$ |
| InceptionV3 | $0.928 \pm 0.001$ | $0.925 \pm 0.005$ | $0.930 \pm 0.005$ | $0.976 \pm 0.003$ | $0.925 \pm 0.005$ | $0.930 \pm 0.005$ |
| Xception | $0.946 \pm 0.007$ | $0.943 \pm 0.008$ | $0.948 \pm 0.010$ | $0.979 \pm 0.004$ | $0.943 \pm 0.008$ | $0.948 \pm 0.010$ |
| InceptionResNetV2 | $0.935 \pm 0.006$ | $0.932 \pm 0.008$ | $0.938 \pm 0.007$ | $0.980 \pm 0.005$ | $0.932 \pm 0.008$ | $0.938 \pm 0.007$ |
| DenseNet121 | $0.956 \pm 0.008$ | $0.948 \pm 0.014$ | $0.965 \pm 0.009$ | $0.990 \pm 0.003$ | $0.948 \pm 0.014$ | $0.965 \pm 0.009$ |
| MobileNetV2 | $0.948 \pm 0.008$ | $0.941 \pm 0.012$ | $0.955 \pm 0.015$ | $0.987 \pm 0.003$ | $0.948 \pm 0.008$ | $0.897 \pm 0.016$ |

## Malaria classification model
### Evaluating pre-trained neural network architectures

The malaria classification models were trained on the NIH dataset. Both the pre-trained neural network VGG16 and VGG19 architectures performed the best, both achieving approximately 0.9600 accuracy and an AUC of at least 0.9900, as shown in Table 3 and Fig. 2. However, we see that VGG16 was slightly less prone to overfitting than VGG19, despite the slightly slower decline in testing loss. In addition, VGG16 required slightly fewer processing cycles to fit a slightly smaller amount of parameters. Consequently, the VGG16 model was selected for further hyperparameter tuning.

Transfer Learning Performance by Architecture

**Figure 2** **CNN performance with different pre-trained architectures.** (A) displays the testing accuracy for pre-trained CNNs. (B) displays the testing loss for pre-trained CNNs.

### Optimizing classification layers

Changing the number of nodes in the two dense layers after the convolution blocks did not affect the final convergence accuracy, as shown in Fig. 3A. However, increasing the number of nodes did allow the model to converge faster. Consequently, 1024 nodes were used for each dense layer during further hyperparameter tuning. A dropout rate of both 0.25 and 0.50 outperformed a dropout rate of 0.75 based on the slightly higher convergence accuracy and faster training. This suggests that a dropout rate of 0.75 may be too heavy of a regularizer. However, the dropout rate of 0.25 began to overfit significantly more than the dropout rate of 0.50. Consequently, a dropout rate of 0.50 was used for each dense layer during further hyperparameter tuning. Lastly, the ReLU activation function appeared to achieve a lower testing loss, compared to the Tanh activation function, so a ReLU activation function was used in subsequent model variants. Visualization of the effects of these hyperparameters on model training is provided in Fig. 3.

### Fine-tuning training hyperparameters

In Figs. 4B–4F, the optimal learning rate for the SGD, RMSProp, Adam, Nadam, and Adamax optimizers are shown to be $10^{-4}$, $10^{-6}$, $10^{-6}$, $10^{-6}$, and $10^{-5}$, respectively. The best learning rates of each optimizer are shown in Fig. 4A, where we see that SGD with Nesterov momentum has the fastest rise to peak accuracy, while maintaining a low testing loss even after convergence. This suggested that SGD with Nesterov momentum with a learning rate of $10^{-5}$ was the best optimizer to move forward with. Meanwhile, Figs. 4G and 4H show that a batch size of 64 provides the fastest convergence while avoiding overfitting.

## Image resolution upscaling

There was a general increase in performance of the FSRCNN model trained on the NIH dataset in terms of PSNR as the number of mapping convolutions ($m$), high-resolution feature dimension ($d$), and low-resolution feature dimension ($s$) increased, as shown in Table 4. The results were derived from the most recent epoch without a dip in testing loss, as some epochs saw a temporary and drastic drop in MSE.

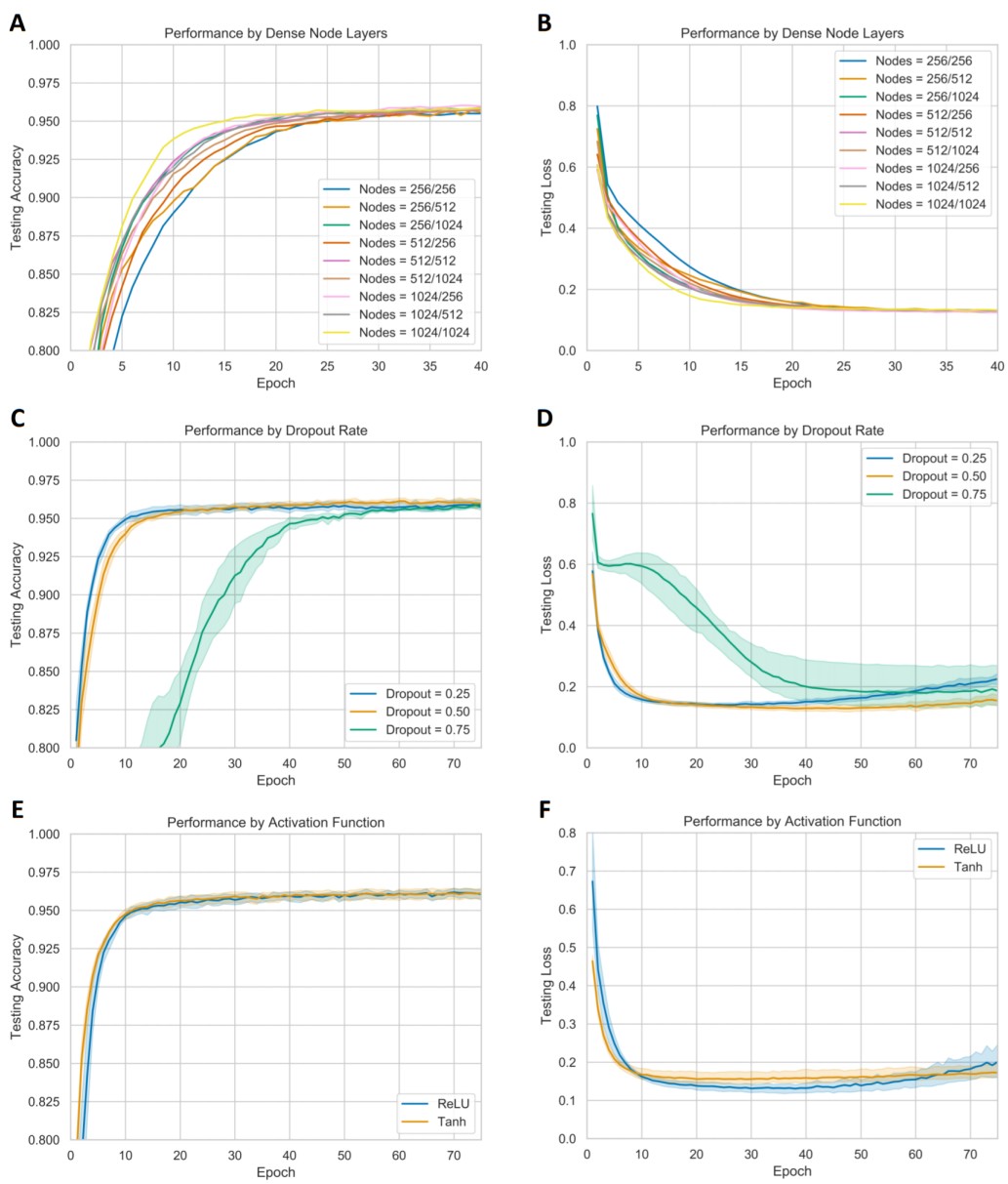

**Figure 3** **Performance of models with different classification layer hyperparameters.** Sections (A–B) display the testing accuracy and loss with different number of nodes in each of the two dense layers. Sections (C–D) display the testing accuracy and loss with different dropout rates after the dense layers. Sections (E–F) display the testing accuracy and loss of the ReLU and Tanh activation functions in the dense layers.

The best performing FSRCNN had a PSNR of 30.79 and a MSE of 54.66. In contrast, the traditional method of bicubic interpolation yielded a PSNR of 24.10 and a MSE of 254.67, as shown in Fig. 5 with sample images. The performance values for the bicubic interpolated images were derived from the entire NIH dataset. In addition, the FSRCNN-derived images

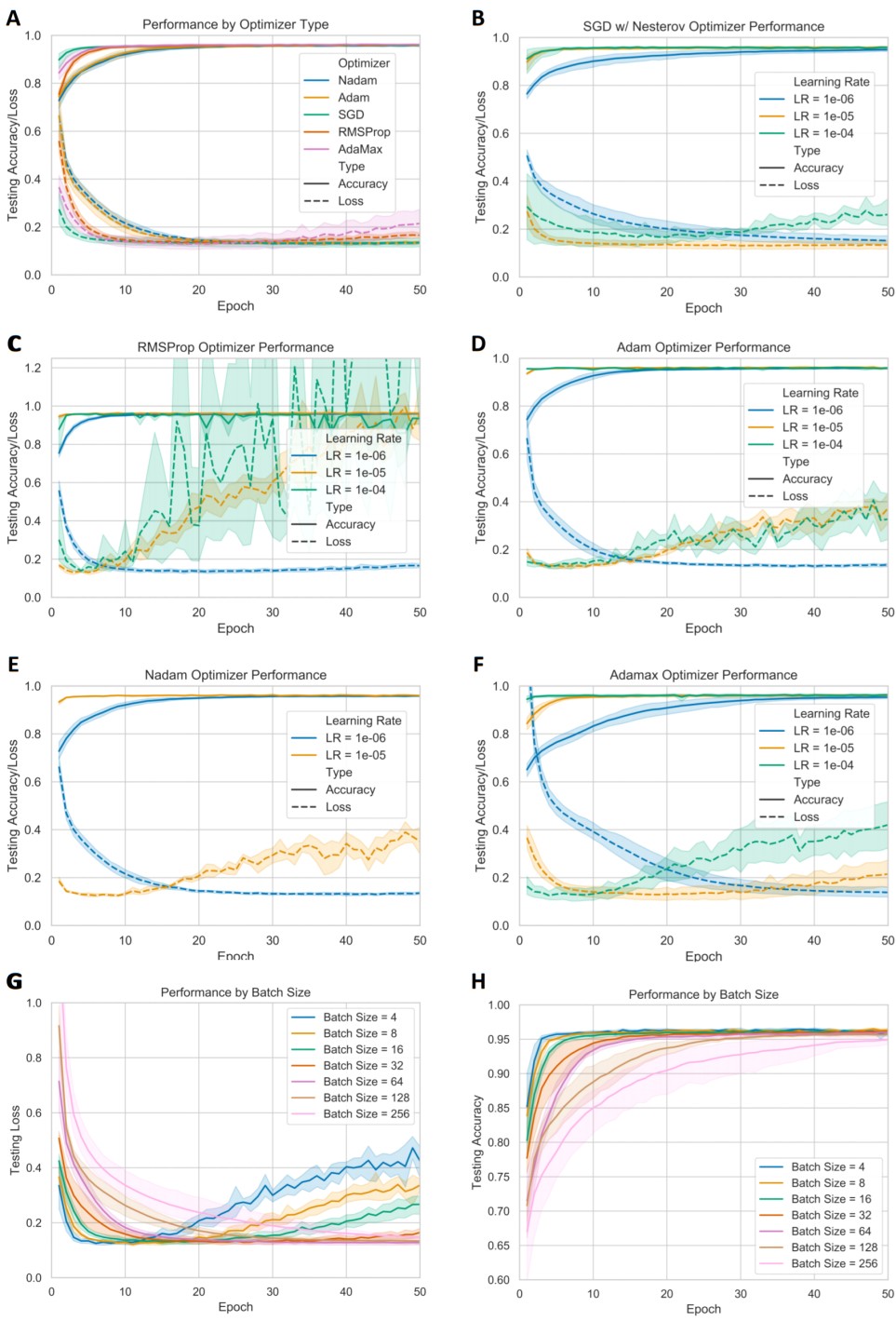

**Figure 4 Performance of models with different optimizers and learning rates.** Section (A) displays the testing accuracy and testing loss of the best performing learning rates of each optimizer, defined as having a fast convergence speed with minimal overfitting. Sections (B–F) displays the testing accuracy and loss of individual optimizers across different learning rates. Results from learning rates that resulted in a lack of improvement were omitted for clarity. Sections (G–H) display the testing loss and testing accuracy across different batch sizes when using a SGD w/ Nesterov optimizer with a learning rate of $10^{-5}$.

**Table 4  PSNR of different FSRCNN variants.** MSE in parenthesis.

| Settings | $m = 2$ | $m = 3$ | $m = 4$ |
|---|---|---|---|
| $d = 48, s = 12$ | 30.09 (64.12) | 30.07 (64.42) | 30.18 (62.85) |
| $d = 48, s = 16$ | 30.30 (61.10) | 30.59 (57.18) | 30.72 (55.53) |
| $d = 56, s = 12$ | 30.10 (64.03) | 30.25 (61.95) | 30.21 (62.39) |
| $d = 56, s = 16$ | 30.42 (59.51) | 30.65 (56.48) | 30.79 (54.66) |

Notes.

$m$, number of mapping layers; $d$, high feature dimension space; $s$, low feature dimension space.

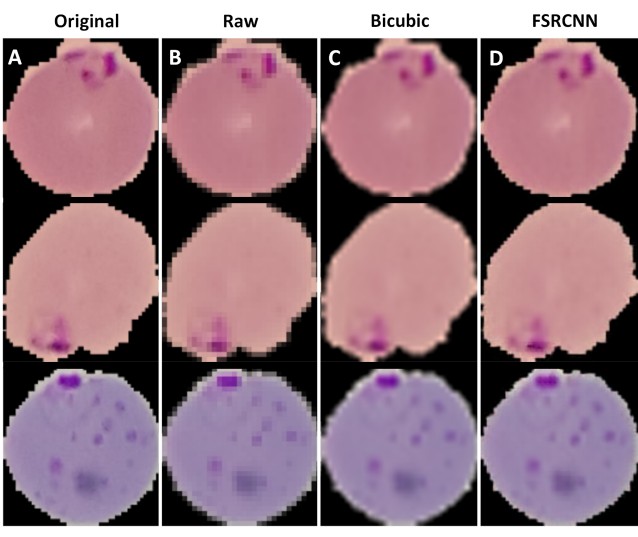

**Figure 5  Sample of resolution enhanced images.** Three individual *P. falciparum*-infected red blood cells from the NIH dataset. Section (A) shows the original $128 \times 128$ pixel images, while Section (B) shows the downscaled $32 \times 32$ pixel images. Section (C) displays the upscaled images via bicubic interpolation and Section (D) displays the upscaled images via the FSRCNN model.

**Table 5  Classification model performance metric with different datasets (mean ± std).** The original dataset contains original $128 \times 128$ images. The FSRCNN and bicubic intepolation datasets consist of downsampled $32 \times 32$ images that were rescaled upwards with their respective methods.

| Dataset | Accuracy | Sensitivity | Specificity | AUC | F1 | MCC |
|---|---|---|---|---|---|---|
| Original High-Resolution | 0.9653 ± 0.0043 | 0.9500 ± 0.0067 | 0.9807 ± 0.0025 | 0.9940 ± 0.0010 | 0.9648 ± 0.0043 | 0.9330 ± 0.0082 |
| FSRCNN | 0.9628 ± 0.0035 | 0.9441 ± 0.0052 | 0.9815 ± 0.0027 | 0.9935 ± 0.0008 | 0.9621 ± 0.0034 | 0.9283 ± 0.0064 |
| Bicubic Interpolation | 0.9486 ± 0.0043 | 0.9093 ± 0.0106 | 0.9878 ± 0.0048 | 0.9913 ± 0.0008 | 0.9464 ± 0.0050 | 0.9022 ± 0.0078 |

were classified more accurately than the raw low-resolution images or bicubic interpolated images in the finalized CNN classification model, as shown in Table 5.

## Integration of CNNs on mobile platform

The Android app takes in an unprocessed photo of a Giemsa-stained thin blood smear, that the user manually selects on the app. Consequently, the image may either be taken directly with the phone camera or electronically acquired through other means. The SSD300 model then isolates individual images of the red blood cells and discard images of white blood

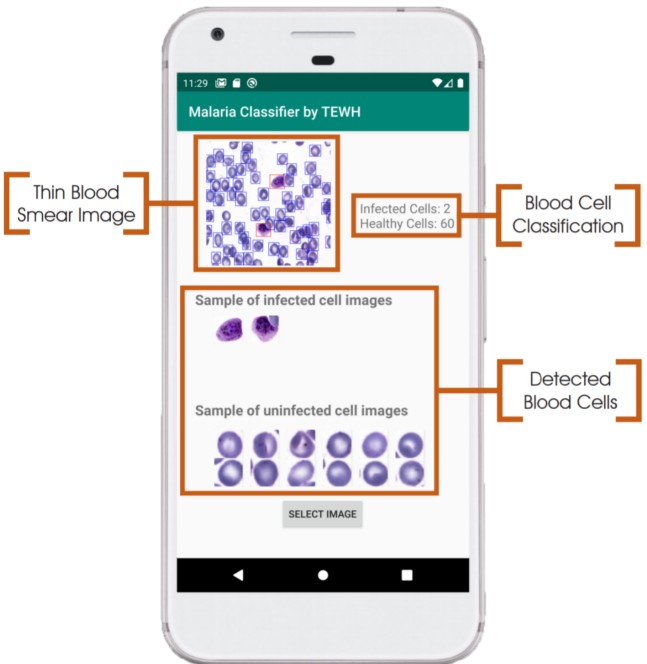

**Figure 6  Example of user interface for malaria screening app.** On the top left is the original thin blood smear image with the object detection bounding boxes overlaid on it. Individual images of red blood cells, as well as cell counts, are also provided.

cells. The image resolution of these individual images is examined to determine whether to upscale the image resolution via the FSRCNN model. Finally, the images are resized to 128 × 128 pixels and run through the VGG16 classification CNN, giving an output indicating the number of uninfected and infected red blood cells, as shown in Fig. 6. Each of the three models is self-contained within .tflite files. Any newly developed model can be similarly exported as a new .tflite file to replace older models. This allows for the mobile app to run different models by only replacing the .tflite files.

# DISCUSSION

## Evaluation of individual deep learning components

The high average precision and relatively low average recall from the SSD300 object detection model indicate that while the detected red blood cells are rarely false positives, a significant portion of red blood cells remain undetected. Because the object detection model does not distinguish between infected and uninfected red blood cells, it is unclear whether one class of red blood cells are more likely to be go undetected by the SSD300 model. However, it would be ideal that both infected and uninfected red blood cells are equally likely to be detected by the object detection model, because the severity of a malaria infection is often measured in percent parasitemia rather than the absolute number of infected red blood cells.

In the FSRCNN image upscaler, we see that while the resolution enhancement process generates significant improvements in the CNN classification model performance, compared to the traditional scaling method bicubic interpolation. This shows that even for simplistic structures such as red blood cells, low-resolution images will cause the classification model to perform significantly more poorly even with traditional image upscaling methods such as bicubic interpolation. This is a critical consideration to keep in mind, as image resolution may be insufficient during the image acquisition process if the camera has poor resolution and the cropped images of individual red blood cells are smaller than $128 \times 128$ pixels. Additionally, we see that increasing the number of mapping layers, the high-resolution feature dimension, and low-resolution feature dimension, all tend to promote an increase in the effectiveness of resolution upscaling. However, it is worth noting that the central purpose of the FSRCNN model is to demonstrate whether improved resolution upscaling methods can positively impact subsequent classification. Recent developments suggest that the use of novel generative adversarial networks (GANs) - such as the SRGAN - yield a better PSNR, and may be better models to implement during further development (*Ledig et al., 2017*).

Meanwhile, our classification CNN model has an accuracy of about 96.53% and an AUC of 0.994, which is lower than the accuracies of other groups who have also trained their model on the NIH dataset. However, it is worth noting that the highest performance reported by *Rajaraman, Jaeger & Antani (2019)* was due to the use of ensemble networks, which may not be feasible for mobile phone use due to its heavier computational burden. Meanwhile, the highest performance reported by *Rahman et al. (2019)* was from a model trained on a modified NIH dataset, in which the group reports that incorrectly labeled images were removed from the dataset prior to training. Top-performing non-ensemble models reported by *Liang et al. (2017)* and *Rajaraman et al. (2018)* report classification accuracies of about 97.4% and 98.6%, respectively. However, neither group tested their final models on a separate independent dataset to examine the generalizability of their models. The performance of our NIH dataset-trained classification model significantly dropped when tested on the Broad Institute dataset, with AUC of $0.945 \pm 0.025$, compared to an AUC of $0.994 \pm 0.001$ with the cross-validated NIH dataset. This suggests that the current classification model is overtrained on the three following differences between the NIH and Broad Institute datasets: (1) unsegmented vs segmented images, (2) *P. falciparum* vs *P. vivax* parasites, and (3) overlapping vs non-overlapping cells in individual images.

## Eliminating the need for internet access and manual segmentation in the mobile app

We present a proof-of-concept with our streamlined, mobile phone-powered screening platform. A flexible Android app framework has been developed, with an easily upgradable modular architecture. Additionally, the code outside of the .tflite files within the Android app is basic and brief, performing basic tasks such as transferring the outputs of the resolution upscaling model to the classification model for diagnostic results. While other groups such as *Rajaraman et al. (2018)* have reported similarly designed mobile phone apps, the apps transmit images to a cloud-based model for classification. This poses an

additional barrier in areas with low or non-existent mobile phone internet connectivity. To our knowledge, our phone app is the only malaria screening app that is currently reported to run entirely on the mobile phone without the need for internet access. In addition, our mobile phone app requires only a thin blood smear image, rather than already segmented images of each individual red blood cell. This removes the need for the technician to manually crop images of each red blood cell to run the single-cell classifier model, a task that is arguably more tedious than the traditional method of classifying each cell manually.

## Immediate barriers to deployment

The two major barriers towards employing the phone-based deep learning models are: (1) the lack of a comprehensive malaria blood smear dataset and (2) the generalizability of the models.

### *Lack of comprehensive dataset*

The NIH dataset contains images of individual *P. falciparum*-infected red blood cells that are already segmented. Meanwhile, the Broad Institute dataset contains images of *P. vivax*-infected red blood cells with bounding boxes but no segmented images. Consequently, this results in a dilemma for realistic application in developing countries. In order to effectively utilize a classification CNN trained on segmented images, we must develop a corresponding cell segmentation model. However, the lack of a dataset with both segmented and unsegmented images makes it impossible to develop such a model. This is problematic for our current models, in which the SSD object detection model was trained for object detection rather than image segmentation, while the classification model was trained on segmented images. Alternatively, a classification CNN could be trained on unsegmented images and only bound images of individual red blood cells, as seen in the Broad Institute dataset. However, the Broad Institute dataset contains *P. vivax* parasites, rather than the predominant and deadlier *P. falciparum* parasites found in African regions. Consequently, an important immediate objective is to acquire a comprehensive dataset that alleviates these issues.

### *Generalizability of deep learning models*

Although *P. falciparum* accounts for the majority of malaria infections in African regions, *P. vivax* is indeed the second most common parasite. In a low-resource setting, it is difficult if not impossible to discern which specific parasite is present in a thin-blood smear outside of manual observation of the thin blood smears. Consequently, an important improvement over current advances would be developing a generalizable deep learning model that is able to indiscriminately detect malaria-infected red blood cells, regardless of the specific parasite present. It seems that no group has attempted this yet. Lastly, as seen in the Broad Institute dataset, there is often significant overlap between individual red blood cells, which may interfere with the accuracy of our current classification model, which was trained on non-overlapping individual red blood cells.

## CONCLUSIONS

While many groups have attempted to use machine learning algorithms to automate the detection and classification of malaria-infected red blood cells, there has not been significant effort towards object detection and image resolution upscaling in the context of the malaria screening process.

By introducing a proof-of-concept, with a preliminary SSD300 object detection model and FSRCNN resolution upscaling model in tandem with a single-cell classification model, we show that a streamlined and sequential approach towards automating the diagnosis of malaria from input of the blood smear to output of the number of infected and uninfected red blood cells may be possible as the individual models are further developed.

With the rapid advancements made every year in deep learning technology, faster and more accurate models developed in the near future can easily be switched with the models used our phone app due to the modularity of our code. This allows us to move closer towards real implementation in developing countries without the need for trained technicians or internet-based computing resources.

## ACKNOWLEDGEMENTS

We would like to thank our anonymous reviewers for their helpful feedback during the revision of this manuscript.

### Funding

The research was funded by donations provided to Texas Engineering World Health (TEWH), a student-chapter of the parent organization Engineering World Health, based at The University of Texas at Austin. Individual donors and other TEWH members that are not listed on the authorship list had no role in any part of the research or writing of the manuscript. There was no additional external funding received for this study. The funders had no role in study design, data collection and analysis, decision to publish, or preparation of the manuscript.

### Grant Disclosures

The following grant information was disclosed by the authors:
The University of Texas at Austin.

### Competing Interests

The authors declare there are no competing interests.

### Author Contributions

- Oliver S. Zhao and Nicholas Chu conceived and designed the experiments, performed the experiments, analyzed the data, prepared figures and/or tables, authored or reviewed drafts of the paper, and approved the final draft.

- Nikhil Kolluri, Anagata Anand conceived and designed the experiments, performed the experiments, prepared figures and/or tables, authored or reviewed drafts of the paper, and approved the final draft.
- Ravali Bhavaraju and Sandhya Tiku performed the experiments, analyzed the data, prepared figures and/or tables, authored or reviewed drafts of the paper, and approved the final draft.
- Aditya Ojha, Dat Nguyen and Ryan Chen performed the experiments, analyzed the data, authored or reviewed drafts of the paper, and approved the final draft.
- Adriane Morales and Deepti Valliappan performed the experiments, authored or reviewed drafts of the paper, and approved the final draft.
- Juhi P. Patel and Kevin Nguyen performed the experiments, analyzed the data, prepared figures and/or tables, and approved the final draft.

## Data Availability

Data is available in GitHub: https://github.com/oliver29063/MalariaDiagnosis. The NIH NLM Dataset is available at https://lhncbc.nlm.nih.gov/publication/pub9932. The Broad Institute Dataset is available at https://data.broadinstitute.org/bbbc/BBBC041/.

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
