# Peer review of "Convolutional neural networks to automate the screening of malaria in low-resource countries"

_PeerJ, doi:10.7717/peerj.9674_

## Round 0.1 · original submission · Minor Revisions

I apologize for not getting this back to you sooner. It happened that I was away from Internet access for 10 days right after the second review was submitted.

The reviewers have provided overall favorable comments, but they have both noted areas for improvement. Most of these are related to the actual manuscript as opposed to the research methodology. Therefore, I have made a "Minor Revisions" decision. Please make sure to address each of their comments carefully. Make sure also that the writing and grammar are accurate and consistent throughout the manuscript.

Reviewer 1 ·

Basic reporting

The authors presented the background very clearly, with up to date statistics. They reviewed widely and provided good summary of previous works on CNNs for malaria detection.

Figure 4 presents too much information, most of which not referenced in the main text. These figures may be presented as supplementary instead.

Experimental design

While the research area is not new, the authors defined a relevant gap and presented a sound machine learning pipeline.

The experiments done were very rigorous and commendable.

While the topic and application of resolution upscaling is interesting and good results were demonstrated, the motivation behind it is not clear in the text. The flow of the narrative would greatly improve if reasons are explicitly stated. Otherwise, the section seems a bit separated from the rest.

The section on the integration to the mobile platform seems insufficient. One of the main ‘selling point’ is to be able to run this system in low-resource countries but very little weight was given to this topic. Perhaps additional implementation details and a review on other CV apps on Android may help strengthen the section.

Validity of the findings

Performance metrics were defined well and although previous works performed better, the authors provided sound reasoning as to why it was the case and why their model is still worth looking into.
Limitations were also stated.

The conclusion involves code modularity but this is not well-argued in the paper. While there are obvious components in the proposed platform, the replaceability of each is not well-established.

·

Basic reporting

1.1 Professional standards of clarity, non-ambiguity, technically correct English

The introduction is very well-crafted English. However, in the methods, results, and discussion sections, there are occasional words missing. In addition, in these sections, there are several instances of subject-verb number disagreement (annotated in PDF).

1.2 Literature references including sufficient field background and context

The methods section is severely under-referenced and does not indicate the sources, either academic or commercial, of most of the algorithms and platforms used. Computing terminology is not consistently explained at a level appropriate for a medical/biological journal.

1.3 Professional article structure, figures, tables

Text is in the correct article structure. Figures and tables are interpolated in the PDF provided to reviewers. Correct procedure is to submit figures and tables as separate files; however, the manuscript PDF provided may have been integrated by the PeerJ content management system. No other issues.

1.4 Self-contained with relevant results to hypotheses.

It should be made clearer in the abstract and introduction that the primary goal is proof-of-concept of a smartphone-executed detection algorithm without need for high-speed internet connection(s). Otherwise no issues.

Experimental design

2.1 Scope
The research is relevant to medical practice and is within scope.

2.2 Research question well defined, relevant and meaningful
The research goal is well defined and shown to be relevant and meaningful. A sharper emphasis on the exact knowledge gap (a smartphone-only based app suitable for field use in Africa) and best applications of the knowledge obtained would be helpful (see comments in PDF).

2.3 Rigorous investigation, high technical and ethical standards
No issues.

2.4 Methods described with sufficient detail & information to replicate
Some ambiguity in which algorithms, and which versions of which algorithms, were used. Insufficient references given in Methods section to fully source all algorithms. In a few cases, ambiguity existed on which dataset was used in which step.

Validity of the findings

3.1 Novelty and replication
Novelty is assessed but only in terms of the goal parameters (development of smartphone-only algorithm). Meaningful replication is encouraged for further development of robustness.

3.2 Data integrity
No issues.

3.3 Sound conclusions
Conclusions do not extend beyond the goal of proof-of-concept, appropriate for the work performed.

3.4 Speculation identified
No issues.

Additional comments

This is a work advancing the field of automated malarial smear reading with the notable step of eliminating the need for cloud-based processing. The authors show acceptable performance within the limits of their training sets; however, they acknowledge that better datasets with wider variety and less prone to overtraining are needed for production of a usable tool in field medical work. The manuscript as currently published has nonstandard reference formatting and insufficient references, particularly in the Methods section. It also has insufficient explanation of computational technical terms to fit the journal’s intended scope. However, the methods and results are solid overall and the topic is appropriate and sufficiently interesting to merit consideration.

---

## Round 0.2 · Minor Revisions

Thank you for addressing the reviewers' comments. I request that you make a few additional changes. Please modify figures 2-4 so that they use a white background (with gray grid lines) rather than a gray background. It's a bit difficult to see the confidence bounds on the lines in the graphs with the gray background. Also, please update the color schemes on these figures to use a colorblind-friend color palette. 8% of men and 0.5% of women are red/green colorblind and thus may not be able to differentiate those lines very well. One option for finding colorblind-friendly palettes is to use colorbrewer2.org. Sorry that I did not notice this earlier.

---

## Author Rebuttal · Round 0.2

**Department of Biomedical Engineering**
The University of Texas at Austin
107 W Dean Keeton St
Austin TX 78712
Phone: +1 (512)-592-1826
Email: oliver.zhao@utexas.edu                                      June 16, 2020

Dear Dr. Piccolo and Reviewers,

We wish to thank the reviewers for their constructive comments on the manuscript and have edited the manuscript to address each point.

In particular, we included more supporting references for many of our models and algorithms in our methodology. In addition, we have also explained technical machine learning terms that may not be well known to the broader PeerJ audience to better fit the scope of PeerJ.

We believe the manuscript is now suitable for publication in PeerJ.

Oliver S. Zhao

On behalf of all authors.

# Reviewer 1 (Anonymous)

*Basic reporting*

*The authors presented the background very clearly, with up to date statistics. They reviewed widely and provided good summary of previous works on CNNs for malaria detection.*

*Figure 4 presents too much information, most of which are not referenced in the main text. These figures may be presented as supplementary instead.*

We agree that Figure 4 presents quite a bit of information. However, we believe Figures 3B-3F are important to illustrate the impacts of the optimizer learning rate on the model performance, while Figure 3G and 3H are also important for the readers to understand the effects of batch size on the model performance. Rather than moving part of Figure 4 to the supplementary section, we believe that it may be better to instead explain these results to more detail in the results section. If the proposed edits on lines 245-247 and lines 250-251 are not recommended, we will happily move part of Figure 4 into the supplementary.

========================================================================

*Experimental design*

*While the research area is not new, the authors defined a relevant gap and presented a sound machine learning pipeline.*

*The experiments done were very rigorous and commendable.*

*While the topic and application of resolution upscaling is interesting and good results are demonstrated, the motivation behind it is not clear in the text. The flow of the narrative would greatly improve if reasons are explicitly stated. Otherwise, the section seems a bit separated from the rest.*

*The section on the integration to the mobile platform seems insufficient. One of the main 'selling point' is to be able to run this system in low-resource countries but very little weight was given to this topic. Perhaps additional implementation details and a review on other CV apps on Android may help strengthen the section.*

We agree that the motivation behind the use of a resolution upscaling model is not clear. To address this, we commented on lines 88-90 that "The FSRCNN model is only utilized if the images of individual red blood cells are of insufficient resolution due to the use of low-end cameras to acquire the thin blood smear images".

We believe part of why the "selling point" of our ability to run our system in low-resource countries is unclear, is due to our poor explanation of the usefulness of our FSRCNN image enhancement model, as indicated in the paragraph above. To our knowledge, we are the only

group to use an image enhancement model for malaria screening. This is important for low-resource countries due to the use of cameras that may not be able to achieve sufficient resolution for our final classification model, which we have indicated in lines 88-90. We believe that by further explaining the purpose of our FSRCNN model, we address this issue.

In addition, in lines 94-97 of our introduction section we state that our platform is efficient enough to run "operate exclusively on the smartphone hardware, eliminating the need for high-speed internet access to transmit image information into a cloud-based neural network model", as many current models are too bulky to run on mobile phone processors. Meanwhile, on lines 317-321, we state that "While other groups such as [Rajaraman, 2018] have reported similarly designed mobile apps, our phone all is the only malaria screening app that is currently reported to run entirely on the mobile phone without the need for internet access", which is critical in low-resource countries that may have spotty or unreliable internet access.

====================================================================

*Validity of the findings*

*Performance metrics were defined well and although previous works performed better, the authors provided sound reasoning as to why it was the case and why their model is still worth looking into.*
*Limitations were also stated.*

*The conclusion provides code modularity but this is not well-argued in the paper. While there are obvious components in the proposed platform, the replaceability of each is not well-established.*

We agree that we did not adequately explain and support our claim that we have high code modularity. To further support this claim, on lines 269-273, we added comments to explain that "Each of the three models are self-contained within .tflite files. Any newly developed model can be similarly exported as a new .tflite file to replace older models. This allows for the mobile app to run different models by only changing the .tflite files".

# Reviewer 2 (Allen Bryan)

## General Comments

*Basic Reporting - 1.1 Professional standards of clarity, non-ambiguity, technically correct English*

*The introduction is very well-crafted English. However, in the methods, results, and discussion sections, there are occasional words missing. In addition, in these sections, there are several instances of subject-verb number disagreement (annotated in PDF).*

Agreed. These missing words and subject-verb number disagreements are indeed present and have been corrected accordingly with the reviewer's helpful highlights in his annotated PDF of our manuscript. Corrections were made at the following lines: 185, 206, 277.

======================================================================

*Basic Reporting - 1.2 Literature references including sufficient field background and context*

*The methods section is severely under-referenced and does not indicate the sources, either academic or commercial, of most of the algorithms and platforms used. Computing terminology is not consistently explained at a level appropriate for a medical/biological journal.*

We agree that the methods were not adequately cited.

To address this, we provided the website links to the two public datasets on lines 103 and 108.

We also indicated the sources of all of the pre-trained CNN architectures used for our classification model on lines 172-176 with the following sources: (Karen Simonyan, 2015), (Kaiming He, 2016), (Chollet, 2016), (Szegedy, 2014), and (Huang, 2016).

We also indicated the sources of all optimization algorithms used in our classification models on lines 188-189 with the following sources: (Kingma, 2014) and (Ruder, 2017).

Lastly, we provided concise explanations of several computing terminologies such as "intersection over union (IoU)" at lines 131-132 and "Matthews correlation coefficient (MCC)" at lines 134-135. In addition, we rewritten parts of our methods section to make certain computing terminology more clear to the broader biomedical audience, such as line 229 where we state that "VGG16 requires significantly fewer processing cycles" instead of "VGG16 requires slightly less operations". In addition, we clarified that the model "accuracy" is "classification accuracy" on line 133.

======================================================================

*Basic Reporting - 1.3 Professional article structure, figures, and tables*

*Text is in the correct article structure. Figures and tables are interpolated in the PDF provided to reviewers. Correct procedure is to submit figures and tables as separate files; however, the manuscript PDF provided may have been integrated by the PeerJ content management system. No other issues.*

We have submitted figures and tables as separate files. The manuscript PDF does indeed contain the interpolated figures and tables due to the PeerJ content management system for LaTeX/Overleaf submissions.

========================================================================

*Basic Reporting - 1.4 Self-contained with relevant results to hypotheses*

*It should be made clearer in the abstract and introduction that the primary goal is a proof-of-concept smartphone-executed detection algorithm without need for high-speed internet connection(s). Otherwise no issues.*

We believe the primary goal is to assist technicians in diagnosis, rather than the lack of need for high-speed internet (which is more of a bonus feature). We elaborated on this on lines 67-74 to make this more clear.

========================================================================

*Experimental Design - 2.1 Scope*

*The research is relevant to medical practice and is within scope*

No comment.

========================================================================

*Experimental Design - 2.2 Research question well defined, relevant and meaningful*

*The research goal is well defined and shown to be relevant and meaningful. A sharper emphasis on the exact knowledge gap (a smartphone-only based app suitable for field use in Africa) and best applications of the knowledge obtained would be helpful (see comments in PDF).*

We agree that a sharper focus on the knowledge gap would be helpful, so we included additional content on lines 67-74 that while there are technicians trained for malaria diagnosis through microscopy, the training is often inadequate and technicians would benefit with assistance from an automated mobile phone screening app.

================================================================

*Experimental Design - 2.3 Rigorous investigation, high technical and ethical standards*

*No issues.*

No comment.

================================================================

*Experimental Design - 2.4 Methods described with sufficient detail & information to replicate*

*Some ambiguity in which algorithms, and which versions of which algorithms, were used. Insufficient references given in the Methods section to fully source all algorithms. In a few cases, ambiguity on which dataset was used in which step.*

We agree that there are insufficient references given in the Methods section to fully source all algorithms, we have addressed this in earlier comments which we display below verbatim for viewing convenience:

> "We agree that the methods were not adequately cited.
>
> To address this, we provided the website links to the two public datasets on lines 103 and 108.
>
> We also indicated the sources of all of the pre-trained CNN architectures used for our classification model on lines 172-176 with the following sources: (Karen Simonyan, 2015), (Kaiming He, 2016), (Chollet, 2016), (Szegedy, 2014), and (Huang, 2016).
>
> We also indicated the sources of all optimization algorithms used in our classification models on lines 188-189 with the following sources: (Kingma, 2014) and (Ruder, 2017)."

With regards to the algorithm versions, while these are not explicitly stated in the manuscript, we believe our Github source code contains the relevant packages and functions used to call each algorithm, which allows an external researcher to easily replicate our experiments.

We agree that there is ambiguity in which dataset was used in certain steps. To address this, on line 225 we clarified that the VGG16 and VGG19 classification models were trained on the NIH dataset. In addition, on line 218 we clarified that the SSD300 object detection model is trained on the Broad Institute dataset. Lastly, on line 253, we clarified that the FSRCNN image enhancement model is trained on the NIH dataset.

================================================================

*Validity of the findings - 3.1 Novelty and replication*

*Novelty is assessed but only in terms of the goal parameters (development of smartphone-only algorithm). Meaningful replication is encouraged for further development of robustness.*

We agree that additional replication of our experiments could provide further robustness in our results. However, it is standard practice in the development of machine learning models to use simple cross-validation rather than repeated cross-validation to gauge the robustness of the machine learning models. In our experiments, we performed 5-fold cross-validation. For example, a previous paper published on PeerJ using neural networks for malaria classification used 5-fold cross-validation as well. [Pre-trained convolutional neural networks as feature extractors toward improved malaria parasite detection in thin blood smear images, Sivaramakrishnan Rajaraman, et al]. We believe that repeating these experiments to general multiple cross-validation sets may be computationally infeasible due to high computing costs.

========================================================================

*Validity of the findings - 3.2 Data integrity*

*No issues.*

No comment.

========================================================================

*Validity of the findings - 3.3 Sound conclusions*

*Conclusions do not extend beyond the goal of proof-of-concept, appropriate for the work performed.*

No comment.

========================================================================

*Validity of the findings - 3.4 Speculation identified*

*No issues.*

No comment.

========================================================================

*Comments for the author*
*This is a work advancing the field of automated malarial smear reading with the notable step of eliminating the need for cloud-based processing. The authors show acceptable performance within the limits of their training sets; however, they acknowledge that better datasets with wider variety and less prone to overtraining are needed for production of a usable tool in field medical work. The manuscript as currently published has nonstandard reference formatting and insufficient references, particularly in the Methods section. It also has insufficient explanation of computational technical terms to fit the journal's intended scope. However, the methods and results are solid overall and the topic is appropriate and sufficiently interesting to merit consideration.*

We agree that the reference formatting was poorly compiled and have recompiled the format. In addition, we explained computational technical terms, as described in earlier comments.

# Reviewer 1 (Allen Bryan)

## Annotated PDF Comments

*P.5 L 19-20 Technically correct English grammar, but not as clear as possible due to confusing order of comments. Suggest instead: "with Africa representing 93% of total cases and 94% of deaths.".*

We agree, we have made the edits accordingly by replacing "with 93% and 94% of total malaria cases and deaths occurring in Africa, respectively", with "with Africa representing 93% of total cases and 94% of total deaths".

======================================================================

*P.6 L 45-46 Again, suggest as per comment in Abstract.*

We have made the edits accordingly.

======================================================================

*P.6 L 61-68 This is a slightly different constraint than the one presented in the abstract. Notice that a technician is still required to prepare and stain the slide and set up the light microscope. Despite not fully meeting the demand, there are a large number of technicians dedicated to malaria diagnostics in sub-Saharan Africa. Suggest a few more references describing the situation for technicians in Africa and phrasing the need in terms of increasing technician productivity while reducing inter-observer variability and aiding training.*

We agree this is a slightly different constraint. We added three additional sources, (Ezeoke, 2012), (Ngasala, 2008), and (Nazare-Pembele, 2016) to support these points, as suggested on lines 67-74.

======================================================================

*P.6 L 77 First area that needs additional background and references. PeerJ is a medical/biological journal. Readers may know of CNNs, but multibox models are out of scope and need more explanation. SSD300 is a specific model published in 2015, and should be referenced either here, in Methods, or both.*

We agree, we added the citation (Liu et al., 2015), who developed the SSD models to make the SSD300 variant more easily accessible to readers.

======================================================================

*P.6 L 79-80 A second algorithm that needs a reference. In addition, there is not explanation of why an image enhancement from low resolution would be a desirable feature.*

We agree, we added an explanation for why the image enhancement model is desirable on lines 81-83 and included the reference (Dong, 2016).

===================================================================

*P.6 L 82 Confusingly worded, suggest "our screening platform uses thin blood smear images as input to provide…"*

We agree the wording is confusing. We changed the wording with the reviewer's suggestion.

===================================================================

*P.6 L 83-84 The term "parasitemia load" or "parasitemia burden" is a shorthand for this concept and should be used after the first description here.*

We adjusted this accordingly on lines 62-63 and lines 93-94.

===================================================================

*P.7 L 90-93 Publicly available datasets not included in Supplementary Material should be referenced with a link. Privately obtained datasets should be acknowledged as privately obtained research material.*

All datasets are public, we included links on these lines for the reader's accessibility/convenience.

===================================================================

*P.7 L 97 Please clarify: all the NIH infections are P. falciparum and all the Broad infections are P. vivax?*

Yes, this is true. We clarified this because we agree that this may not be clear to readers.

===================================================================

*P.7 L 99 I don't care if you reference this as a product (needing a reference to provider, in this case Google, Inc., Mountain View, CA) or as an implementation of algorithms (in which case, scholarly references describing its operation are required). But it needs to have SOME indication of source.*

We agree that we did not reference the GCP appropriately. We added the provider reference as (Google LLC, Mountainview, CA). Note that Google has recently restructured from Google Inc. to Google LLC.

================================================================

*P.7 L 100 When were each configuration used?*

================================================================

*P.7 L 100-102 Are these specifications of actual hardware or virtual platforms? What does "N1 machine" mean? To a computer engineer, these terms are clearly options in Google Cloud, but this is not so clear to medical and biological researchers.*

N1 machine is a configuration for the Google Cloud Platform. We agree this may be confusing to non-computational medical and biological researchers. We have written the formal name "N1 high memory machine" instead of the colloquial term "N1 machine" so that the term can be easily located online and be well-defined.

================================================================

*P.7 L 103 Used to do what? Again, you and I know what a boost disk does and what Debian is, but the average reader will not.*

We agree this may be confusing to the average reader. On line 118 we added new text saying "... to run all software on the Google Cloud Platform" to explain to the reader that the boot disk and Debian is the platform used to run our code. We believe this clarifies enough detail, without inundating the reader with additional details.

================================================================

*P.7 L 114 It would deeply help reader understanding to note the equivalent terms in medical research, as these terms have different meanings in different fields. Avg. precision corresponds to positive predictive value and average recall corresponds to sensitivity.*

We agree this would be helpful. We added a brief explanation of this on lines 136-138.

================================================================

*P.7 L 115 Why this metric? Nonobvious.*

We agree it is nonobvious. We explained why IoU is used in lines 131-132 to address this.

================================================================

*P.7 L 116 Accuracy is a term with varying definitions across fields; suggest being specific in description.*

We agree accuracy has an ambiguous definition. We clarified that it is "classification accuracy", which is the percentage of correct classifications for both classes.

========================================================================

*P.7 L 117 Unless you have a reference count limit, a reference here wouldn't hurt. MCC is the same thing as the phi statistic, but only in bioinformatics is the term MCC used frequently.*

We agree. We included a reference and also adjusted our text to indicate that the term is equivalent to the phi statistic for non-bioinformatics readers.

========================================================================

*P.8 L 121 A repository for which you will provide the link in either references or supplementary material, yes?*

Yes, this is indeed true.

========================================================================

*P.8 L 123 It is absolutely imperative to cite a reference here.*

We added the (Liu et al., 2015) reference to address the lack of reference here.

========================================================================

*P.8 L 129 Also needs a reference!*

We added the (Ruder, 2016) reference to address this comment.

========================================================================

*P.8 L 131 Scaled up or down? Or both? Lossless or lossy scaling? What algorithm? What image format?*

We agree that this line was not explained clearly. We adjusted it to indicate that the images are scaled down to 300x300 through bilinear interpolation (the latter detail which makes it implicit that it is lossy scaling).

========================================================================

*P.8 L 135 Why the much smaller size than the 300x300 above? For ease of processing on smartphone hardware? For requirements of the algorithm?*

The 300x300 size is for the entire thin blood smear image, which the images of individual red blood cells is 128x128. It is not strictly for the algorithm requirements, in which we can adjust the image input size requirements. However, the traditional SSD300 (as the name suggests) is trained on 300x300 size images, so we thought it would be best to maintain this size. The image sizes are not for the purpose of ease of processing either (although the size does have an impact).

=====================================================================

*P.8 L 149-150 Does He et al. describe or reference descriptions of all the rest of the models?*

He et al. describes a portion, but not all of the models. We have included new references for readers to locate the other models mentioned in these lines.

=====================================================================

*P.8 L 161 had yet (subject-verb number agreement)*

We adjusted this accordingly.

=====================================================================

*P.8 L 163-164 Again, references? Or list as products and give manufacturers?*

We included the new sources (Kingma and Ba, 2014) and (Ruder, 2016), which contains information on all of the optimizers mentioned.

=====================================================================

*P.9 L 172-184 You can get away with switching from past to present tense to mark off the internal workings of the FSRCNN model. But once you move outside the algorithm in the second paragraph, you MUST switch back to past tense, as these are actions done by the researchers, not the algorithms.*

We adjusted the tenses within the algorithm and in the second paragraph, as suggested by the reviewer.

=====================================================================

*P.9 L 186 Reference?*

We added a reference accordingly.

=====================================================================

*P.9 L 195 You MUST explain the importance of IoU for your readers for these results to be interpretable.*

We agree, we added text in the methods section on lines 131-132 to explain the importance of IoU to a broader audience.

=====================================================================

*P.9 L 202 On which dataset? Since you have two, must needs be clear whether you are testing on one, the other, or both. Don't assume that because you mentioned that in Methods that it goes without saying here.*

We agree, we added text on line 225 to indicate that the classification models were trained on the NIH dataset.

=====================================================================

*P.9 L 205 Jargon not familiar to med/bio audience. Suggest "slightly fewer processing cycles" or some such.*

Agreed. We have adjusted as suggested.

=====================================================================

*P.10 Table 2 and Figure 1 Suggest explaining AP and AR abbreviations once, in the table caption, to avoid repetition repeats of repetition.*

We agree, we changed the description in Table 2 to make it clear that AP stands for average precision and AR stands for average recall.

=====================================================================

*P.10 Table 3 From which dataset? Don't make the reader go back to Methods to find out when it only takes a few words.*

We agree that it makes more sense to just explicitly state which dataset was used. We adjusted the description for Table 3 to indicate that it is the NIH dataset.

==========================================================================

*P.11 L 208-209 Where are the data that show this?*

This is shown in Figure 3A. To make this more clear, we added text at line 247 to explicitly state this so that the reader can be directed to the data more easily.

==========================================================================

*P.11 L 215-216 Even I don't know what you mean here. "Testing loss"? Activation of what?*

Testing loss is a technical term for measuring the error in the model predictions (opposed to the more colloquial term "testing loss"). An activation function is a nonlinear function that is applied to the weighted inputs at a given node in a neural network. We believe "testing loss" and "activation function" is a well-defined and well-known term for those who are familiar with neural networks.

==========================================================================

*P.14 L 214 indicate. Two subjects, precision and recall, so plural form.*

We have fixed this accordingly.

==========================================================================

*P.14 L 214 Missing a verb!*

We have fixed this accordingly.

==========================================================================

*P.16 L 249 See, this should be said in the introduction.*

We agree that this should have been explained in the introduction instead. We included new text on lines 61-63 and explained that severity is measured as percent parasitemia.

==========================================================================

*P.16 252-253 Not exactly a surprising result when expressed this way. What matters is HOW low-resolution you can go without notable performance loss.*

We agree that this statement was not expressed the best way. At lines 287-288 we included the new text saying "This shows that even for simplistic structures… low-resolution image upscaling methods will cause the classification model to perform significantly more poorly, even with traditional image upscaling methods such as bicubic interpolation".

========================================================================

*P.16 L 255 A vague term! How poor?*

We agree that the term for "poor" was vague. We clarified on line 289 and line 290 where we state that "if the camera has poor resolution and the cropped images of individual red blood cells are smaller than 128x128 pixels".

========================================================================

*P.16 L 270 neither group*

We have adjusted this subject-verb number agreement grammatical error accordingly.

========================================================================

*P.16 L 274 towards what?*

We adjusted the text here to more clearly state that "This suggests that the current classification model is overtrained on the three following differences…" rather than vaguely saying "This suggests that the current classification model does not generalize well towards due [sic] to the three following".

========================================================================

*P.16 L 278 This is the real breakthrough component of the paper and should receive more emphasis.*

Agreed, we emphasized this more in lines 322-324.

========================================================================

*P.16 L 279-280 "with an easily upgradable modular architecture" says this much more succinctly.*

Agreed. Changed as suggested.

===================================================================

*P.16 L 285-288 Highlight this more!!*

We explained more in-depth in lines 322-324 that our streamlined process is more efficient than previous classifiers because previous classifiers can only take in images of individual red blood cells. This means a technician using the machine learning app must do this manually, which is arguably even more tedious than the traditional method of manually classifying each cell. Thus, our 3-step process removes the barrier to make the automated method more efficient than manual examination by the technician.

===================================================================

*P.18 L 347-350 You reference all these datasets in the text without letting the reader know where the links are. Say something like "(see Data Availability)" the first time you refer to each of these so that reader isn't left hanging!*

We recognize this may be confusing when the datasets are first referred in the methods section. Consequently, we included the links in the methods section as well, as seen in lines 103, 108, and 142.

===================================================================

*P.18 L 357 What kind of reference format is this?? Strongly recommend rechecking.*

Apologies, we realized the LaTeX .bib file we used was not formatted correctly. We have adjusted it to be

===================================================================

*P.18 L 358 Oh no, you can't just say "It's on arXiv". Give a resource number!*

We agree on this comment. We have adjusted the citations to include the arXiv web ID number. For example, the preprint by Diederik P. Kingma "Adam: A Method for Stochastic Optimization" has a uniquely identifiable ID as "arXiv.1412.6980".

===================================================================

*P.18 L 359 Capitalize: "United States". You wouldn't write "in france and spain", would you?*

We have fixed this accordingly.

===================================================================

*P.19 L 418 ePubs still need a reference indicator of some sort to verify that you have the right paper. A DOI number will do.*

We have adjusted the citation for this accordingly and included the DOI number.

===========================================================================

**End of Document.**

---

## Round 0.3 · Minor Revisions

Thanks for updating those figures. I am so sorry, but I realized there is one more issue that needs to be addressed. The code is available in GitHub, but there is no README page or instructions on how to run the code. I would like to test your code to make sure it runs properly. Please update the GitHub site so that it is clear what needs to be done. What software (and versions) need to be installed? How is it executed, etc.

---

## Round 0.4 · accepted · Accept

The GitHub repo looks great. Thanks and congratulations!